# (Non)Marketing of Breastmilk Substitutes in South African Parenting Magazines: How Marketing Regulations May Be Working

**DOI:** 10.3390/ijerph19106050

**Published:** 2022-05-16

**Authors:** Sara Jewett, Sukoluhle Pilime, Linda Richter

**Affiliations:** 1Division of Health & Society, School of Public Health, University of the Witwatersrand, Johannesburg 2193, South Africa; sppilime@gmail.com; 2DSI-NRF Centre of Excellence in Human Development, University of the Witwatersrand, Johannesburg 2193, South Africa; linda.richter@wits.ac.za

**Keywords:** breastmilk substitutes, breastfeeding, the WHO code for the marketing of breastmilk substitutes, regulations, magazines, South Africa, sleeping, crying, posseting

## Abstract

Although exclusive breastfeeding (EBF) for the first six months is optimal for child health, it remains low globally. Breastmilk substitutes (BMS) marketing undermines breastfeeding. In 2012, South Africa introduced Regulation 991, which prohibits marketing BMS products for infants below 6 months. Our study aimed to explore if and how BMS products were presented in South African parenting magazines post-R991. We applied a mixed-methods cross-sectional content analysis design, analyzing all 2018 issues of two popular parenting magazines. We descriptively analyzed quantitative codes, derived from an a priori framework, and conducted qualitative content analysis on a subset of texts and images. We found there was no overt marketing of BMS to parents with infants below 6 months. However, BMS advertisements were placed next to articles about young infants, and vague wording and images were ways by which BMS companies may indirectly benefit. Medical experts in both magazines promoted the introduction of solids before six months. To conclude, while BMS companies in South Africa were abiding by R991 by not overtly advertising BMS in parental print media, their influence persists. Continued monitoring of print media as well as other channels is advisable. This study may be of interest to countries considering stronger regulations of BMS advertising.

## 1. Introduction

According to the World Health Organization (WHO), breastfeeding is the best source of nutrition for the healthy growth and development of infants [1,2], with significant positive health benefits to breastfeeding mothers as well [3]. Exclusive breastfeeding (EBF) is defined as giving only breastmilk without any other liquids such as water or herbal preparations, with the exception of vitamins, mineral supplements or medicines, in the first six months of life [4]. Despite its superiority as an infant feeding method, the global weighted prevalence of EBF in 57 low and middle income countries (LMIC) was only 45.7% between 2000 and 2018 [5]. In South Africa, the most recent national statistics suggests 32% of infants under six months are exclusively breastfed, with only 24% of those aged 4–5 months still exclusively breastfeeding [6].

Infant crying, sleeping and posseting behaviors are among the many factors known to undermine breastfeeding [7]. Posseting is a feeding condition in which infants regurgitate or ‘spit up’ milk, sometimes confused with gastro-esophageal reflux (GER) or the more serious gastro-esophageal reflux disease (GERD) [8]. Problems with excessive crying, sleeping and feeding are reported in approximately 20% of infants [9], with high levels of co-morbidity noted between posseting, crying and sleeping interruptions [10]. Parents of infants experiencing these issues frequently change feeding methods in their efforts to soothe their children [9]. A systematic review of infant feeding influences in South Africa noted how the pressure from family to respond to these infant cues has consistently been associated with the early introduction of formula and solids over three decades [11]. The same review reported an increasing “culture” of formula use.

In an effort to promote breastfeeding, the International Code for Marketing of Breast-Milk Substitutes (‘the Code’) was adopted by the Thirty-third World Health Assembly in 1981 [12] and was signed by South Africa. The Code was developed to counteract the marketing of breastmilk substitutes (BMS), stipulating contexts in which marketing should be restricted. According to Article 5, the advertising, promoting and provision of BMS to women, mothers or members of their families directly or indirectly should not be undertaken. The Code’s definition of BMS includes commercial infant formula, other milk products marketed for children up to 36 months, foods and beverages for infants younger than 6 months and any other foods or beverages represented to be suitable for use as a partial/total replacement of breastmilk, including feeding bottles and teats [12] (p. 8).

The role of the commercial formula industry in undermining EBF is well established and has been elaborated on in a recent WHO/UNICEF joint report [13] about how marketing violations influence infant feeding. Despite the clear definitions contained in the Code, BMS companies continue to use overt and covert means to market their products throughout the world [14], including through the sponsorship of pediatric associations [15] and through the use of social media [16,17]. For instance, scholars have recently drawn attention to the increased use of social influencers in Latin American countries which have had Code legislation for decades [17], pointing to changing marketing strategies as opposed to a decline in or cessation of marketing. Similarly, the use of social media to market BMS has recently been flagged as a new avenue being used by the industry in the South African context [18]. In particular, social media influencers [19], who have access to a large audience and have the ability to persuade other people’s decisions, are increasingly used by companies producing BMS to promote and market their products [20].

Despite South Africa’s adoption of Regulation 991 to enforce the Code [21], local violations have been reported [22], including in magazines [23]. While there has been substantial research about overt BMS advertising in parenting magazines in high income countries [24,25,26,27,28,29], similar research in LMIC contexts is more limited and not frequently published in the peer-reviewed literature [23]. A quantitative content analysis of ‘hand feeding’ vs. breastfeeding advertisements in a popular parenting magazine in the United States from 1972 to 2000 found high levels of advertisements for formula, solids/cereals or hand feeding equipment: 249 examples in 84 issues [28]. Furthermore, they found that the frequency of advertisements in a given year negatively predicted breastfeeding among new mothers in the following year. A more recent analysis of formula advertising from 2007 to 2012 in the United States found continued high rates of advertising, with an aggressive upward trend from 2009 [27]. However, such findings may not be relevant to South Africa, as the US is not a signatory of the Code and does not regulate BMS advertising. However, even in Taiwan, a country with local regulations to enforce the Code, high frequencies of BMS advertisements were found in parenting magazines [29].

In addition to not knowing the frequency of BMS advertising in South African parenting magazines, we were unclear about which marketing tactics are being used in this medium. The tactic of ‘pain point’ marketing, where real or perceived problems are highlighted by manufacturers, with their products presented as the solution, was highlighted as a BMS marketing tactic in the recent 2022 report by WHO and UNICEF. The ‘pain’ would be construed as breastmilk or a BMS company’s competitors (other brands) not meeting the needs of their prospective consumer (in this case, parents). A related tactic is focusing on the science of BMS products to address such issues [13]. For example, the BMS industry has created a wide range of special formulations and stand-alone thickeners to address posseting [30], despite a systematic review of such products finding no benefits for breastfed infants [31]. Whether the BMS industry is using tactics such as pain points and scientific claims to promote their products in South African magazines was unknown at the onset of this study.

We were interested in whether local regulations may be acting as a deterrent for overt marketing in magazines and wanted to explore whether BMS companies are employing more covert means of marketing their products. Our research questions were: Are BMS products being marketed in popular parenting magazines in South Africa in the context of R991? If so, how are they marketed? In this study, we reviewed popular South African parenting magazines with the aim of exploring if and how BMS products were presented in the context of infants crying, sleeping and/or posseting (parental pain points).

What follows is a presentation of our study methodology, followed by our quantitative results and qualitative analysis. In the conclusion, we summarize the key results and discuss these in the context of the global literature to draw out implications, both for countries with regulations, such as South Africa, as well as for countries that may be considering regulations. We also discuss the limitations of this study and recommend directions for future research.

## 2. Methods

### 2.1. Design

We applied a mixed-methods cross-sectional media content analysis design to address our study aim. Media content analysis is a non-intrusive descriptive methodology focused on the characteristics and form of message content as well as making inferences about content creators and/or audiences [32]. This methodology can be applied to multiple forms of media, which in our case was print magazines. While some methodologists, such as Neuendorf, argue that media content analysis should be primarily quantitative, we applied the form of media content analysis grounded in the behaviorist tradition defined by Shoemaker and Reese [33], as described in Macnamara [32]. Specifically, they argued that in addition to quantitative descriptions, there is a qualitative element to media content analysis, which enables researchers to infer what effect media content might produce in terms of opinions, attitudes and perceptions through deeper qualitative analysis of textual and visual content.

In our case, we were interested not only in whether BMS was marketed in the contexts of sleep, crying and/or posseting, which could be answered purely quantitatively, but also in the question of how BMS was marketed, for which both quantitative and qualitative analysis is needed. The second question was of particular interest, as this could facilitate inferences on how messaging might influence the feeding choices of parents exposed to the magazines. Qualitatively, we used textual analysis, drawing on semiotics, to look at how signs or signals within the media, such as images of a happily sleeping baby near an advertisement for BMS, might influence audiences [32].

### 2.2. Study Population and Sample

The study population was popular print magazines targeting parents in South Africa. Two magazines were purposively sampled for this study based on their wide readership and differences in audience. Your Baby (YB) and Mamas and Papas (MP) were selected as being the two parenting magazines with the highest readership [34] out of the five in circulation. YB is represented by Parent24, South Africa’s largest parenting website, which has independent Twitter, Facebook, Pinterest and Instagram handles. MP is distinguished by its targeting of parents from “diversified racial groups” and its efforts to “fuse the old and new methods of parenting” [35]. Only print issues were included.

Ethical approval was not required, as the study did not involve humans and drew on publicly accessible media content. The rationale for not counting this as human research is explained in a review of ethical considerations for social media research [36]. Nevertheless, we opted to exclude the names of the individuals who we quoted, as our intent was not to draw attention to individuals but rather to patterns in the text and images analyzed. This is in line with the ethical principle of respect of persons.

For this research, we analyzed all issues of the 2018 print magazines for YB (6 issues) and MP (12 issues). The decision to focus on a single year was guided by the larger study within which this study was nested. The World Health Organization had commissioned a number of studies globally to explore, through multiple methods, the ways that the BMS industry, and, more specifically, commercial infant formula companies, might be violating the Code.

Within these issues, all written content that addressed sleeping, crying or posseting in relation to infant feeding (formula, breastmilk or otherwise) was deemed eligible. Similarly, any images portraying infant feeding, sleeping, crying or posseting were included in data extraction to enable qualitative semiotic analysis. The rationale for including forms of feeding beyond BMS was to enable a balanced comparison of the media content (text and images) that parents were exposed to, a strategy recommended by Macnamara [32].

### 2.3. Data Collection

For message identification and quantitative analysis, we used an a priori design, including codes to establish a consistent coding framework. Each variable included a definition to support the systematization of coding. Two coders worked with the initial coding systems to identify categories to be included under each variable and to refine definitions. Once these were defined, we applied Mayring’s [37] approach of matching the a priori categories to the text (vs. text to category) to increase reliability. Specific variables were developed to explore the presence of BMS messages or images in relation to written content, as well as the slant for text content (positive, neutral or negative) and sources of messages. The content type, e.g., letter, advertisement or feature, was noted for context. Coders also typed verbatim quotes from the text to enable qualitative textual analysis at a later stage, which could be accessed alongside the actual print magazines. See Appendix A for the text coding framework.

The data capturing tool was managed in Microsoft Excel. Two researchers independently coded each magazine issue in separate spreadsheets, which were later compared. Where there was a difference in coding, a third coder was brought in to independently code, and inconsistencies were discussed within the team to develop a consensus. Once all magazines were double-coded, the inter-coder reliability of each variable was calculated manually for percent agreement. One variable entitled “reality” was removed for being too inconsistently applied. On all other variables, over 80% agreement was attained. The first author (SJ) made final decisions about the codes used for analysis. These final codes were imported into STATA 16.1 for analysis. Of non-explicit BMS claims that were initially coded, those that did not contain any specific claims about the three conditions of interest were removed during data cleaning, all of which were breastfeeding articles. The final dataset is available from the first author on request.

### 2.4. Analysis

Simple descriptive analysis was used to quantify the presence of overt BMS advertisements and to characterize their contextual relationship to articles or images related to crying, sleeping and/or posseting. Mentions of crying, sleeping and/or posseting in relation to more vague feeding practices were also included in the analysis. We included coded text on other BMS claims in the analysis for potential relevance to R991 violations. Any questionable behavior, such as concealing age limits in product images, was referred by the authors to the Department of Health (DoH) through their official email codewatch@health.gov.za. Other practices, such as promoting early feeding before six months, were also referred to the DoH.

As a simple description of frequencies was insufficient to explore our second question of how BMS products were potentially being marketed. Additional qualitative textual analysis was conducted by the first author on a sub-set of texts purposively selected for their (1) representativeness, (2) disconfirming examples and (3) exceptional examples to illustrate the range of messages we encountered. Those selected as representative were identified from frequent patterns we observed, e.g., expert advice to introduce solids from the age of 4 months. Disconfirming examples were selected by looking at differences in slants or claims. The reporting of disconfirming examples increases the credibility of this analysis, as they guard against bias. Finally, exceptional examples were those that, though infrequent, were still highly relevant to the research questions, such as overt BMS product marketing. The same process of sampling was followed with images that were embedded in or next to articles, using the same rationale. Given the complexity of considering how both text and images may work together, hard copies of the magazines were used to support this analysis. The use of thick descriptive analysis, including verbatim quotes, was used support the analysis. These examples are integrated into the presentation of the quantitative results.

In terms of structuring our analysis, we began by seeking to answer first whether there was any overt or covert advertising of BMS, providing examples to justify our quantification of such. We then moved on to analysis of article and image content, seeking to identify the narrative or semiotic techniques used to present infant feeding. Finally, we focused on claims about BMS specific to the marketing pain points to parents of crying, sleeping or posseting, aligned with our study aim.

## 3. Results

### 3.1. Overt BMS Advertising

None of the 2018 magazine issues overtly advertised that commercial BMS products should be used to feed infants under six months. However, YB (Nov/Dec, p. 77) contained an advertisement for Similac on the same page in which a nurse was quoted as explaining, ‘A common sleep problem in the toddler years is getting up a few times each night to fulfil your little sprog’s demands for endless bottles of milk or juice’. The Similac advertisement on the same page purported to provide a solution to sleep disruption. Unlike the quote by the nurse, the advertisement did not define ages. It read:

If your child is underweight or recovering from illness and needs some catch-up growth, it’s a good idea to give him a specialised nutritional supplement milk *such as* Similac Kid (R144) at bedtime in place of his milk. This will ensure that he is getting quality nutrition at bedtime and *allay any fears you may have of him needing milk feeds during the night*.(*italics added*, p. 77)

The lack of precision about which Similac product was recommended through the phrase “such as” and the suggestion that replacing milk with a BMS product can increase sleep duration and promote recovery represents a form of marketing to a wider audience, known as cross-promotion [13].

As with the Similac example, we identified commercial food and drink advertisements that could undermine EBF. For instance, the YB Sept/Oct edition promoted fruit and vegetable juices and purees alongside articles advising parents about how to improve the nutrition and health of their babies and toddlers. One advertisement read: “Squish 100% Fruit + Vegetable purees and juices have been created with *babies* and toddlers in mind!” (*italics added*, p. 25). While there was a “6 months plus” symbol visible in one advert image, the age was cut off of in most of the product images over a four-page spread (pp. 22–25). The concealing of age advice was also observed in a Purity advertisement in the MP April edition, where the cereal was placed below a picture of a peacefully sleeping infant in his mother’s arms (p. 31). In contrast, another Squish advertisement in MP (Dec 2017/Jan 2018, p. 45) had clear labeling regarding suitability for children older than six months, and there was no counter-narrative in the articles. These examples highlight why it is important to include product placement and images in content analysis.

### 3.2. Narratives Undermining EBF

Both magazines contained narratives that undermined the government’s promotion of EBF for six months by advising parents to introduce solids between 4–6 months. Sometimes, these were not linked to advertisements. For example, one menu for green apple and pea puree was labelled as being appropriate “from 4 months” (YB, Mar/Apr, p. 63). These articles were often positioned near commercial BMS and baby food advertisements. One MP article highlighted that “research shows that you should start your child on solids when they are 6 months old, but in all honesty, it really depends on your baby and when they’re ready”, (May, p. 34) with a Purity baby food advertisement directly below. A South African dietician quoted in YB provided similar advice. She cited the WHO recommendation of 6 months but also quoted contradictory advisories from the American Academy of Allergy, Asthma & Immunology, the European Society for Paediatric Gastroenterology, Hepatology and Nutrition and the Australasian Society of Clinical Immunology and Allergy, concluding that “there is no real definite answer” about when best to introduce solid foods (Mar/Apr, p. 54). In the same issue as the Similac advertisement, a medical expert advised parents that “the appropriate age for introducing solids is between four and six months of age” (YB, Sept/Oct, p. 48). These examples highlight how content can undermine South Africa’s EBF promotion guidelines, with or without direct links to overt BMS advertisements.

### 3.3. Crying, Sleeping and/or Posseting and BMS

In addition to the overt marketing of BMS products, the coding structure enabled us to explore sub-texts of BMS promotion that were embedded in text or imagery in relation to our three conditions of interest (crying, sleeping and posseting) as well as other BMS claims. In total, 46 articles/advertisements were identified and analyzed, 72% (n = 33) of which came from Your Baby.

Table 1 presents a summary of coding results. In 18 articles, claims were made about BMS and at least one of the three conditions of interest, comprising 39% of all coded text. Another 11 articles included claims about infant feeding and at least one of the conditions, without an explicit mention of BMS products. Finally, 17 articles mentioned BMS outside of the context of the three conditions.

Of the three conditions discussed in the context of BMS, crying was the most common focus (43%, n = 20), followed by sleeping (40%, n = 17) and lastly posseting (7%, n = 3). Half of these articles (n = 9) contained claims about BMS and both crying and sleeping. Most of the content came from feature articles or columns (46%), followed by ‘how to’ advice articles, full-page advertisements and finally letters from readers. The prominence of BMS varied by the type of claim, with sleep getting the most attention and posseting, most often framed as “reflux”, receiving the least attention. With the exception of BMS product advertisements, BMS mentions were embedded within broader articles, taking up less than one sentence per paragraph.

In addition to frequency and prominence, we also explored the slant or way in which BMS was discussed. BMS was mostly discussed in a positive or neutral way, with only 14% of the coded text presenting BMS in a negative way. Of the three conditions, BMS and sleep were discussed in the most positive way, followed by crying. Neutral ways of introducing BMS came through phrases such as “When he is feeding on the breast or the bottle…” (YB, Nov/Dec, p. 30). There were other articles that reassured parents using formula that their choices should not be judged, such as an article about the first hour after birth that noted: “Not all women want to breastfeed either, and this does not make her a bad mother.” (MP, May, p. 32).

The way in which the three conditions were discussed had a strong medical bias, focusing on infant health, the nature of the conditions and infant development. In the context of health, there were no references to premature babies, although the idea of a baby ‘growing well’ was raised sometimes, without defining what that meant (see later quote). There was also an empathetic focus on parents, particularly mothers, and attending to their needs. For crying and sleep conditions, there was a focus on how BMS could alleviate the conditions. The exception was a myth-busting article about changing formulas for babies with colic, where the author wrote, “it’s completely hazardous if and when parents self-treat their babies with formula milk…” (YB, July/Aug, p. 41). The topic of BMS cost was not highlighted, except in a side comment that “breastmilk is also a lot cheaper than formula” in an article encouraging fathers to support breastfeeding (YB, Jan/Feb, p. 23). When there were quotes, the most common sources were non-medical experts, such as sleep specialists (n = 16), followed by medical experts (n = 10) and mothers (n = 7).

We explored medical expert sources more deeply in terms of infant feeding advice and our conditions of interest, given their known influence on infant feeding decision-making. While many offered balanced descriptions of what to expect in terms of sleeping and crying at different ages, some experts were more prescriptive. For instance, in an article authored by a nurse about setting routines for babies, she advised that, from six weeks,

…as long as your baby is growing well, do not give your baby milk if less than two and a half hours have elapsed since her last feed. Stretch her as close to three to four hours between feeds as you can by offering her 30 to 50 mL of cooled, boiled water from a spoon or bottle. (YB, Jan/Feb, p. 47)

The same nurse suggested that infants at four months who do not sleep eight hours per night should be introduced to solid foods, with the caveat that parents should talk to their healthcare practitioner. In an article about colic by ‘Dr Dad’, parents were advised that

certain formulas are developed specifically for babies with allergies or reflux, so speak to your doctor or pharmacist for help in choosing an alternative formula for your baby if they are fussy, cramping or have reflux. (MP, July, p. 37)

Specific BMS brands were not mentioned by these experts, though in some cases, there were advertisements in other parts of the same issue.

## 4. Discussion

By 2018, the formula industry seemed to be overtly playing by the rules with regard to direct advertisement of BMS products in line with R991 requirements. That said, we identified concerning practices, such as the concealment of age information on images of BMS products and a potential example of cross-promotion by Similac. We also identified narratives in both magazines, supported by medical experts that encouraged the introduction of solids before six months. The marketing pain points of sleeping and crying had the most assertive claims in relation to BMS product use in the context of slant. South Africa is progressive to have promulgated R991 [21] together with a mechanism to report violations. South Africa is categorized as ‘substantially aligned’ with the Code [38]. Even so, it is unclear whether BMS companies reported for violations are penalized for R991 violations. Having weak mechanisms to enforce the Code is a well-documented barrier to addressing violations [39], one that is being exploited by BMS companies in LMICs [13]. Other countries with regulations attribute similar violations to poor legal enforcement [29], and the lack of sanctions was identified by the WHO as a key weakness, even among countries with legislation [38].

The importance of monitoring Code violations has been emphasized as an important strategy to limit the influence of BMS companies [39]. However, ongoing surveillance of all potential channels used by BMS companies is difficult without the active participation of communities. Part of enforcement, therefore, requires deeper engagement with communities and key stakeholders, such as universities, to increase awareness of both the benefits of breastfeeding as well as unethical BMS marketing practices [39,40]. Indeed, a recent scoping review about Code enforcement in South Africa [40] highlights that a focus on regulation violations is only one part of a multi-pronged strategy needed to address breastfeeding inequities.

This leads us to discuss the less obvious ways that BMS companies may be influencing breastfeeding in parenting magazines. Experts, including those with medical qualifications, were the main sources of quotations in the articles we coded. In some cases, these professionals cast doubt on the WHO EBF recommendations in their advice to parents about when to introduce solids. Other times, they explicitly promoted early feeding. In doing so, they drew on outdated recommendations of introducing solids between 4–6 months, which the WHO updated to 6 months in 2001 [11]. South Africa updated infant feeding advice in 2012 to confirm that EBF for 6 months was optimal for all infants, inclusive of those who were HIV-exposed [11]. This raises serious questions as to why South African medical practitioners are promoting outdated and contradictory advice.

These contradictions are particularly concerning considering that parents refer to health professionals as trusted sources of information for infant feeding [10,11]. Despite relatively strong legislation in place about how BMS companies can engage with the health profession [38], researchers in South Africa have documented how BMS companies are actively seeking to influence health professionals through conferences and the cultivation of professional relationships, leading to conflicts of interest [41]. This tactic was also identified in the UNICEF/WHO report on global trends [13]. The South African study [41] provided specific examples of how BMS companies emphasize to medical professionals how their products help with issues like ‘excessive crying’ and ‘sleepless infants’. Pointing parents to specialized formulas, such as those mentioned by Dr Dad in our earlier example, is exactly what BMS companies intend for medical experts to do, as it reinforces their ‘pain-point’ marketing strategies.

Of the three conditions we explored, sleeping and crying jumped out as being most strongly discussed in the context of BMS, particularly in the context of health. A focus on infant health was also noted in studies of BMS advertising in U.S. and Taiwanese parenting magazines [24,29], suggesting that this may be a common tactic. While the U.S. study identified that most BMS health claims focused on child development or issues such as allergies [24], our own study found high frequencies of health frames focused on sleeping and crying. While pain point marketing strategies tend to focus on issues existing customers are facing, they also encourage businesses to look for the unmet needs of prospective customers, e.g., parents of breastfed infants, including deep insights gained through ethnographic research [42]. Unlike other types of products using pain point techniques, once a BMS company convinces a caregiver to switch from breastfeeding to formula to address real or perceived sleep and crying issues, a return to breastfeeding (relactation) is more difficult than switching between BMS products.

In addition to expert opinions, we noted a trend towards reassuring parents that were facing challenges with crying, sleeping and posseting that their feeding choices were not being judged, including their use of BMS. The pressure individual women experience to be a ‘good mother’ is well documented [10,43]. Infant feeding choices are influenced by a host of factors; individual mothers should not be blamed for not breastfeeding, as it may be a difficult option for some. Nevertheless, we know that the BMS industry has capitalized on this, literally, by selling reassurance as a way to normalize their product and position it as equivalent to breastmilk [13].

While this study sheds light on how BMS was presented in the context of sleep, crying and posseting in magazines, there are limitations. Our analysis focused on two of the most popular parenting magazines in 2018, but by 2021, Mamas and Papas had ceased publishing, highlighting the volatility of the industry. There are numerous parenting magazines in circulation in South Africa that may be making different editorial decisions in relation to their relationships with BMS companies. A recent survey found that 9% of South African women listed magazines as a top source of formula adverting [13], pointing to their continued relevance both to parents and the BMS industry. Additionally, we only analyzed the print forms of these publications, while they also had online and social media formats. In addition to differences in how content may be displayed between print and alternative formats, how the audience engages with that content may be mediated by the formats themselves, according to Neuendorf [44]. As such, our findings only apply to the print format readership. With this in mind, we created our coding framework in such a way that studies can be replicated for print as well as for other forms of media in other contexts.

In terms of methods, our inter-coder reliability measure, percentage agreement, could arguably inflate the coding agreement figures [45]. However, we believe that the inclusion of a coding reliability process to guide our decisions as well as our inclusion of all codes in our ICR measurement (vs. a sample) increases the likelihood of the trustworthiness of our findings. Finally, as we did not engage in primary research, we cannot infer the intentions of the magazine editors and content producers, nor the audience’s interpretation of the content we analyzed. These aspects would need to be researched independently through qualitative research with editors, audience reception research and/or ecological study designs that compare media exposure and behavioral outcomes, such as a study conducted in the United States [28]. We also recommend that studies similar to our own be conducted with other forms of media, particularly social media, given the growing concerns that this is the new frontier of BMS marketing [13,17].

## 5. Conclusions

The role that regulations are playing in curtailing the overt marketing of BMS to infants under 6 months in South African parenting magazines is evident in the absence of clear company violations of R991. Countries that are considering similar regulations can look to South Africa as an example of how such regulations may be acting as a deterrent. However, this does not mean BMS marketing has stopped. Rather, our deeper analysis of text and images is suggestive of BMS companies’ influence on editorial decisions about product placement as well as the advice from medical professionals selected. We looked at a cross-section of print magazines across one year, while evidence abounds that BMS companies are exploiting a range of other channels to influence infant feeding choices [11,17,18,20,36]. As we say in South Africa, ‘a luta continua’ in the quest for optimal infant nutrition.

## Figures and Tables

**Table 1 ijerph-19-06050-t001:** Content analysis of text in two parenting magazines, by claim type and overall.

Content Analysis	Crying % (n)	Sleeping % (n)	Posseting % (n)	Other % (n)	Overall % (n)
**Feeding claim**	**(20)**	**(17)**	**(3)**	**(17)**	**(46)**
BMS claims	65 (13)	65 (11)	67 (2)	0 (0)	39 (18)
Non-BMS claims	35 (7)	35 (6)	33 (1)	0 (0)	24 (11)
Other BMS claims	0 (0)	0 (0)	0 (0)	100 (17)	37 (17)
**Article type**	**(20)**	**(17)**	**(3)**	**(17)**	**(46)**
Feature/column	40 (8)	29 (5)	33 (1)	59 (10)	46 (21)
Letter/discussion	10 (2)	12 (2)	67 (2)	6 (1)	11 (5)
Advice (‘How to’)	20 (4)	24 (4)	0 (0)	24 (4)	24 (11)
Full-page Adverts	30 (6)	35 (6)	0 (0)	12 (2)	20 (9)
**BMS prominence ***	**(13)**	**(11)**	**(2)**	**(17)**	**(35)**
Full article/advert	38 (5)	55 (6)	0 (0)	53 (9)	43 (15)
>1 sentence/paragraph	15 (2)	9 (1)	50 (1)	6 (1)	11 (4)
<1 sentence/paragraph	46 (6)	36 (4)	50 (1)	41 (7)	46 (16)
**Slant towards BMS ***	**(14)**	**(14)**	**(3)**	**(14)**	**(35)**
Positive	43 (6)	64 (9)	0 (0)	71 (10)	54 (19)
Negative	7 (1)	7 (1)	0 (0)	29 (4)	14 (5)
Neutral	50 (7)	29 (4)	100 (3)	0 (0)	31 (11)
**Thematic focus**	**(20)**	**(17)**	**(3)**	**(17)**	**(46)**
Infant health	75 (15)	76 (13)	66 (2)	59 (10)	65 (30)
Nature of condition	70 (14)	82 (14)	33 (1)	6 (1)	54 (25)
Mother well-being	25 (5)	35 (6)	66 (2)	24 (4)	41 (19)
Parenting	30 (6)	29 (5)	0 (0)	24 (4)	28 (13)
Infant development	25 (5)	35 (6)	66 (2)	18 (3)	26 (12)
How BMS contributes	40 (8)	41 (7)	0 (0)	0 (0)	20 (9)
Condition prevalence	15 (3)	18 (3)	0 (0)	0 (0)	9 (4)
Economic/Practical	0 (0)	0 (0)	0 (0)	6 (1)	2 (1)
Unclear (in passing)	35 (7)	41 (7)	33 (1)	18 (3)	33 (15)

* Excluded data where breastmilk substitutes (BMS) were not mentioned, as not relevant.

## Data Availability

The final dataset is available from the corresponding author on request. The raw data are not available on an online repository due to their format (print magazines). The original magazines, on the basis of which the codes were generated, would need to be procured independently.

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
