# Peer review of "(Non)Marketing of Breastmilk Substitutes in South African Parenting Magazines: How Marketing Regulations May Be Working"

_ijerph, 2022, doi:10.3390/ijerph19106050_

Round 1

Reviewer 1 Report

The paper raises a very interesting question of breastfeeding important for early child development. But there are several questions and issues to clarify.

First of all, why cross section data for year 2018 are analyzed even if the regulation was introduced in 2012?

My second concern is related to the fact that most the variables in this research are in the form of codes. How would it be possible to analyze these coding variable? Only in terms of bar charts or pie charts, without for example sample averages and etc? Therefore, I think the paper is more of a qualitative character, especially given the style of results being presented.

Thirdly, the methodology is not described. Just citing two papers in line 89 on page 2 is not sufficient.

Fourthly, no data on babies are analyzed. Information for example about health or weight of babies could also have been important.

Before being able to evaluate I would need the comments above be addressed.

Author Response

Please find in our attached document a point-by-point response to your valuable suggestions. 

Reviewer 2 Report

Abstract: The abstract is too long. It should be reduced by adding the aim of the paper, the used method, and the achieved results.

Introduction: The aim of the paper should be better clarified. Moreover, at the end of the Section, the structure of the paper should be provided.

Section 2: A research question guiding the study should be added. Moreover the paragraph “2.5. Ethical considerations” is too short I suggest removing it and moving the content within the paragraph “Section 2.2 population and sample”

Conclusion: The conclusion should be improved. Moreover, limitations of the study and future studies should be added.

Author Response

(The authors gave the same response as above.)

Reviewer 3 Report

Introduction: Could include some information on whether other countries have implemented regulations similar to regulation 991. Costa Rica is one country that has had similar regulation for a long time. Could also include a short overview of the findings related to BMS advertising in other countries.

Method: 

Line 138: Expand on "Additional qualitative textual analysis was conducted by the first author on a sub-set of texts purposively selected for their 1) representativeness, 2) disconfirming examples, and 3) exceptional examples to illustrate the range of messages we encountered." to illustrate how you arrived at results in 3.1, 3.2 and 3.3.

Results: Table 1 is a little confusing. Perhaps include %, (n) with each column heading and put the bolded numbers in ( ) as they are counts I think.

Line 237: Rethink "When there were quotes, the most common sources were non-medical experts, such as sleep specialists (n=16), medical experts (n=10) and mothers (n=7)." Perhaps adding 'followed by' after (n=16).

Line 246: This line "…do not give your baby milk if less than two and a half hours have elapsed since her last feed. Stretch her as close to three to four hours between feeds as you can by offering her 30 to 50ml of cooled, boiled water from a spoon or bottle. (YB, Jan/Feb, p.47). " is an important issue that could be further discussed in the discussion as it deals with handing out medical advice based on very old feeding recommendations from the 1960s.

Discussion: Start off with a paragraph on the main findings and then build off that.

Line 259: Don't use this so early on in discussion. Perhaps belongs in methods. "Any questionable behavior, such as covering up age limits in product images, was referred by authors to the Department of Health (DoH) through their official email codewatch@health.gov.za. Other practices, such as promoting early feeding before six months, were also referred to the DoH." 

Missing how the current research results compare to other literature on code violations or national regulations.

Author Response

(The authors gave the same response as above.)

Round 2

Reviewer 1 Report

Text in lines 16-19 attempts to address my major comment on the data being used. But it is confusing and not informative. I am not able to understand what exactly data and how they were analyzed.

What local magazines were analyzed? Only 46 observations, this is too few. No reliable research or conclusions can be made based on such a small sample.

Methodologically, the paper does not contribute anything if it is replicating the McNamara article.

Many details are omitted which does not allow me to evaluate the quality of research data or methods. I must suggest rejection.

Reviewer 3 Report

My comments have been addressed.

Author Response

Thank you for reviewing our manuscript a second time. Your first round of comments assisted us in improving it substantially. We have noted that you are satisfied with how we addressed your comments.